# Constraining local ocean dynamic sea-level projections using observations

Dewi Le Bars[1], Iris Keizer[1], Sybren Drijfhout[1,2]

[1]R&D Weather and Climate Modelling, Royal Netherlands Meteorological Institute (KNMI), De Bilt, the Netherlands
[2]Institute for Marine and Atmospheric Research Utrecht, Utrecht University, Utrecht, the Netherlands

*Correspondence to*: Dewi Le Bars (dewi.le.bars@knmi.nl)

**Abstract.** The redistribution of ocean water volume under ocean-atmosphere dynamical processes results in sea-level changes. This process, called Ocean Dynamic Sea Level (ODSL) change, is expected to be one of the main contributors to sea-level rise along the western European coast in the coming decades. State-of-the-art climate model ensembles are used to
make 21[st] century projections for this process, but there is a large model spread. Here, we use the Netherlands as a case study and show that ODSL rate of change for the period 1993-2021 correlates significantly with ODSL anomaly at the end of the century and can therefore be used to constrain projections. Given the difficulty to estimate ODSL changes from observations on the continental shelf, we use three different methods providing seven observational estimates. Despite the broad range of observational estimates, we find that 4 to 16 CMIP6 models have rates above the observational range and 0 to 1 below. We
compare 4 model selection methods which differ on the way the uncertainty in the rate estimation is considered. We find that for stricter selection methods the rate of ODSL is closer to the observational range and the uncertainty in future change is reduced. The influence of model selection is largest for the low emission scenario SSP1-2.6. We discuss the advantages and disadvantages of those selection methods and their suitability to different users.

## 1 Introduction

Understanding local sea-level rise and providing reliable sea-level projections is an important duty of the scientific community to help society face this challenge (Le Cozannet et al. 2017; Hinkel et al. 2019). Currently, sea-level projections use the contributor-based and process-based approaches (Fox-Kemper et al. 2021). Contributor-based means that the projections are the sum of each individual contributors to sea-level rise and process-based means that when possible the contributors are projected using models of the detailed physical processes (Church et al. 2013; Le Bars 2018). The
contributors considered in projections are: glaciers, ice sheets, land water storage, glacial isostatic adjustment from the last glacial maximum, global steric sea level and ocean dynamic sea level (ODSL). ODSL is defined as: "the local height of the sea surface above the geoid with the inverse barometer correction applied" (Gregory et al. 2019). Changes in ODSL are due to changes in winds and ocean currents as well as changes in atmosphere/ocean heat and freshwater fluxes. It is related to both steric and manometric sea-level changes. The contribution of ODSL to local sea-level rise is modelled directly by
coupled atmosphere-ocean general circulation models (AOGCMs, Gregory et al. 2019). Therefore, AOGCMs from the coupled model intercomparison projects 5 and 6 (CMIP5 and CMIP6) were the base for ODSL projections from the

intergovernmental panel on climate change assessment reports 5 (AR5, Church et al. 2013) and 6 (AR6, (Fox-Kemper et al. 2021).

By definition, ODSL change has a global mean of zero. As a result, it contributes to sea-level rise in some areas and to sea-level drop in other areas. From AOGCMs we expect that ODSL is an important contributor to sea-level rise in the coastal North Atlantic and Arctic oceans (Lyu, Zhang, and Church 2020). For the North Sea, ODSL is even expected to be one of the major contributors to sea-level rise during the 21$^{st}$ century (Vries, Katsman, and Drijfhout 2014; Bulgin et al. 2023). In that region, it was also shown that this process is related to changes in the Atlantic Meridional Overturning Circulation
(AMOC) and is larger in CMIP6 than in CMIP5 (Jesse, Le Bars, and Drijfhout 2024). Despite continuous improvement in our understanding and modelling of ODSL (Lyu, Zhang, and Church 2020), there is still a large divergence between the projection of different AOGCMs. For the North Sea, this divergence has even increased in CMIP6 compared to CMIP5 (Jesse, Le Bars, and Drijfhout 2024). The model spread is usually interpreted as a difficulty to predict future sea-level changes resulting in an uncertainty in sea-level projections (Fox-Kemper et al. 2021). Methods have been developed for
other sea-level contributors to constrain projections with observations. For example, sea-level highstands from the paleoclimate archive and recent observations have been used to constrain future Antarctic mass loss (DeConto et al. 2021; van der Linden et al. 2023). Changes in global steric sea level were also constrained using observed ocean temperature changes during the Argo period (Lyu, Zhang, and Church 2021). However, ODSL projections have not yet been constrained by observations. In this study, we develop a method to do so.

Lyu, Zhang, and Church (2021) used observed rates of ocean heat content change and steric sea-level change to constrain future steric sea-level change from CMIP6 models with the method of emergent constraints (Hall et al. 2019). Inspired by this study we explore the use of past ODSL rates to constrain future ODSL from CMIP6 models. We first explore the relation within the CMIP5 and CMIP6 model ensembles between past rates of ODSL for different periods and the ODSL
height anomaly at the end of the century. Since wind forcing has a large influence on inter-annual to inter-decadal variability of ODSL in the North Sea (Keizer et al. 2023), we also analyze the results of CMIP6 models with wind influence on ODSL removed. ODSL is a quantity that was defined to be easily retrieved from AOGCM but not from observations. ODSL can't be measured directly, however it can be estimated from observations. Here we use three ways to estimate ODSL changes. First, we use a method similar to computing a sea-level budget (Frederikse et al. 2016; 2020; Camargo et al. 2023) but
instead of checking if the budget is closed we assume that the budget is closed, treat ODSL as the unknown, and solve the budget equation to find it. Second, we compute the steric sea-level change around the continental shelf in locations of deep ocean assuming that this anomaly is transported to the coast (Bingham and Hughes 2012; Hughes et al. 2019). Third, we use the results of an ocean reanalysis that does not assimilate satellite altimetry data. This provides a range of estimates that we use to select plausible CMIP6 models and compute new ODSL projections. Finally, we discuss the limitations of the CMIP
ensemble to simulate ODSL and of our method to constrain projections.

## 2 Data and method

We use three different methods to estimate ODSL change along the Dutch coast for the period 1993-2021. Those are presented in the first three sections. The analysis of CMIP6 models used for projections is then presented in the following 2 sections. The analysis of all datasets, models and observations, is performed on yearly averaged data. This removes the seasonal cycle and high-frequency variability that are not the focus of this study.

**Steric sea-level change**

To compute the steric influence on sea level along the coast of the Netherlands we first compute ocean water density from quality-controlled ocean temperature and salinity data from the EN4.2.2 dataset (Good, Martin, and Rayner 2013) with the bias correction from Gouretski and Reseghetti (2010) and also from the IAP dataset (Cheng et al. 2017). We use the Thermodynamic Equation of Seawater 2010 (TEOS-10, Millero et al. (2008)). The density is then integrated vertically from the ocean surface down to 2000 m in the extended Bay of Biscay and in the Norwegian Sea (Fig. 1a). This calculation is based on the assumption that because the North Sea is shallow, steric expansion there does not have a significant impact on sea-level change but steric anomalies in the deep ocean propagate to the North Sea as a mass inflow and influence local manometric sea level (Landerer, Jungclaus, and Marotzke 2007). This is equivalent to assuming no horizontal pressure gradient (Bingham and Hughes 2012). The steric anomaly propagation could be either through coastal trapped waves (Calafat, Chambers, and Tsimplis 2012) or other physical processes like internal waves, tidal pumping, eddies or Ekman transports (Huthnance et al. 2022). Previous studies have used the region of the extended Bay of Biscay based on a good multi-year to multi-decadal correlation with observed sea level (Frederikse et al. 2016; Bult et al. 2024). However, it is not clear if that same region is also useful for long term trends, as considered here, therefore we also consider the Norwegian Sea (Fig. 1a). From the regional steric sea level we remove global steric sea level from Frederikse et al. (2020) to obtain an estimate of ODSL change.

**Sea-level budget closure**

Another way to estimate ODSL is to consider it as the unknown in the sea-level budget. We develop two budgets for the coast of the Netherlands. The first one is based on geocentric sea-level observations from satellite altimetry data averaged over a region close to the coast (polygon in Fig. 1b). The second one is based on relative sea-level observations from the 6 reference tide gauges (Vlissingen, Hoek van Holland, IJmuiden, Den Helder, Harlingen, Delfzijl) distributed along the Dutch coast (Keizer et al. 2023). We use ice sheets, glaciers, land water storage and global steric contributions from the budget of Frederikse et al. (2020) which considers gravitation, rotation and viscoelastic deformation effects for all contributions except for global steric sea-level change. Since this budget stops in 2018 we extrapolate the contributions up to 2021 using a linear fit to the last 10 years of the individual time series. This is possible because those terms are rather smooth and because at the inter-annual time scale, local sea-level change in the North Sea is mostly set by wind (Keizer et al. 2023) and regional steric

anomalies (Frederikse et al. 2016). We also include glacial isostatic adjustment from the ICE-6G(VM5a) model (Peltier, Argus, and Drummond 2015). The direct influence of the nodal cycle is assumed to be in equilibrium with the astronomical forcing and is calculated as in Woodworth (2012), which was shown to be a good method when the nodal cycle influences on steric effects are considered separately, as we do here (Bult et al. 2024). Once all known sources of sea-level change above are computed, they are removed from the observed sea level and the effect of wind and inverse barometer on sea level are computed from a multi-linear regression to zonal and meridional wind and pressure fields from the ERA5 reanalysis with the same method as Frederikse et al. (2016).

**Ocean reanalysis**

Ocean model reanalyses, that assimilate observations of temperature and salinity in a dynamical ocean model, also provide an estimate of ODSL. In ocean reanalyses, the relation between the deep ocean and the shelf is computed in a physically consistent manner with the drawback that there is no global ocean reanalysis product yet available that have both the physical mechanisms (e.g. tides) and horizontal resolution necessary to compute the transition between the deep ocean and the shelf (Holt et al. 2017). Some models also assimilate data from satellite altimetry which includes the influence of other contributors on sea level and makes it difficult to know if the output of the reanalysis is ODSL or geocentric sea level. We use here the Simple Ocean Data Assimilation (SODA3.4.2, Carton, Chepurin, and Chen 2018)) which does not assimilate altimetry data. The data covers the period 1980 to 2020 and has a resolution of 0.5°x0.5° and 50 vertical levels on Mercator grid. Atmospheric surface forcing is from ERA-interim and the COARE4 bulk formula is used. To make sure to obtain ODSL from the model output the global mean sea level is removed for each year. The wind influence is also removed using a multi-linear regression between ODSL and zonal and meridional wind from ERA-interim.

**ODSL from CMIP5 and CMIP6 models**

Changes in ODSL are available from the output of the models taking part in CMIP5 and CMIP6 with the variable "zos" but need to be postprocessed. We use here the same data as Jesse, Le Bars, and Drijfhout (2024). The "zos variable has three dimensions: time, latitude and longitude. Four post-processing steps are applied: First, we compute the yearly average from the monthly data. Second, we compute the linear temporal drift in the piControl simulations of each model and remove it from the historical and future scenario simulations for each grid box. This relies on the assumption that the drift is not sensitive to the external forcing (Hobbs, Palmer, and Monselesan 2016). Third, since all models discretise the ocean on different grids, the data is regridded to a common grid. We choose a regular 1°x1° grid. The re-gridding is performed in a computationally efficient way by using the open-source library xESMF with a bilinear method for most models and a nearest-neighbour method for the few models for which the bilinear approach does not work. Additionally, since the land/sea mask is different between models we perform a spatial extrapolation of the available data to where there is no data. This makes sure that all models have data on the same areas. Fourth, we remove the global mean. For CMIP6 models we also

remove the influence of local wind on sea level along the coast of the Netherlands to reduce the influence of natural variability on our results.

To estimate the influence of natural variability on the linear trend over 29 years, we use the historical simulations of each model between 1850 and 1980. We assume that before 1980 the anthropogenic trend is not dominating the trend. Using those 131 years, linear trends are computed starting each year. The standard deviation of those 102 trends is used as an indication of the influence of natural variability.

**Wind influence on ODSL from CMIP6 models**

The wind influence on ODSL from climate models is computed with a multi-linear regression as for satellite and tide gauges data. However, only the zonal and meridional wind are used in the regression. The atmospheric pressure is not used because the zos variable of climate models does not include the inverse barometer effect (Gregory et al. 2019). Wind and ODSL are selected in a region along the Dutch coast (Fig. 1b). To avoid issues with long term trend influencing the regression coefficients, we include a linear trend in the regression model and determine the regression coefficients only on the historical

period. The assumption that the trend is linear does not hold for the combination of historical and scenario period but over the historical period it is reasonable. We then assume that the coefficients relating zonal and meridional wind constituents to ODSL obtained during the historical period also apply to the scenario period. More details about the method and analysis of the results can be found in Keizer (2022).

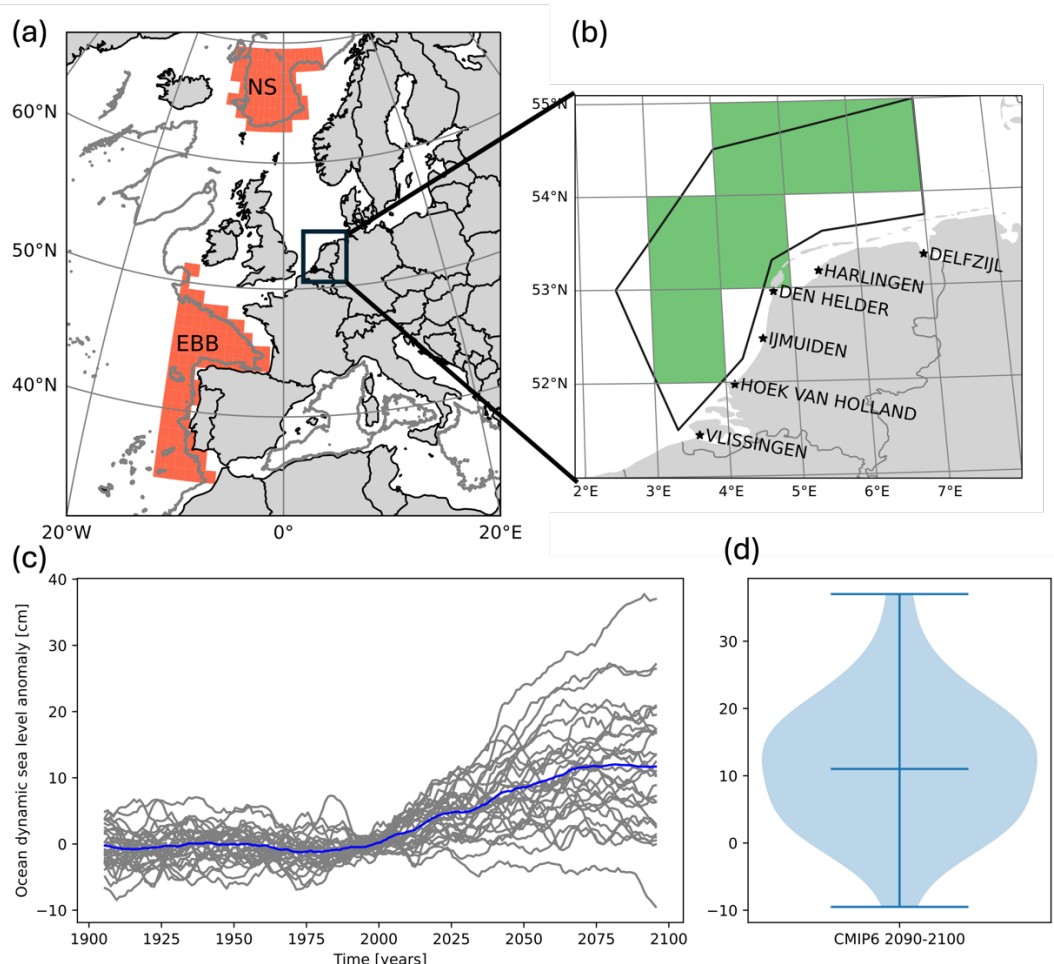

Figure 1: (a) 2000m isobath and the two regions of steric sea-level change computation: Norwegian Sea (NS) and extended Bay of Biscay (EBB). (b) Location and name of the 6 reference tide gauges along the Dutch coast. Horizontal grid (1°x1°) on which the CMIP5 and CMIP6 model data is interpolated and the 6 grid boxes used to compute local ODSL (green). Region used to compute sea-level rise from satellite altimetry and the SODA ocean reanalysis (black polygon). (c) Changes in ODSL for 29 CMIP6 models under the SSP1-2.6 scenario with the reference period 1986-2005 and a 10-year running average applied. (d) Violin plot showing the distribution of ODSL values averaged over the period 2090-2099 for the same models shown in (c).

## 3 Results

In Fig. 1 c,d we show the projections of ODSL for the SSP1-2.6 emission scenario of the CMIP6 models along the coast of the Netherlands. There is a large spread between climate models, even for this low emission scenario. For the average of 2090-2099 the values go from -9.5 cm for the FGOALS-g3 model to 37 cm for CIESM. We now investigate if observed ODSL rates of change can be used as a constraint for future ODSL changes. To do that, we look at the relation between, on the one hand, the rate of sea-level rise in the recent past or near future and, on the other hand, the sea-level change between the end of the century 2090-2099 and the reference period 1986-2005 (Fig. 2). We computed the rate of sea-level rise for

periods between 15 and 50 years (y-axis) ending between 1960 and 2050 for both CMIP5 (top row) and CMIP6 (middle row) models. We see that for rates computed over shorter period the correlation coefficient depends more strongly on the end date of the period than for rates computed over longer periods. This is especially the case for the CMIP5 ensemble with positive correlations for periods ending in the 1980[th] followed by negative correlations for periods ending around 2000 and again positive correlation for periods ending later. Removing wind influence on sea level (3[rd] row Fig. 2) has a limited influence on reducing the variability in the correlation. For the CMIP6 ensemble the correlations are higher for the low emission scenarios (SSP1-2.6, SSP2-4.5) than for the high emission scenario (SSP5-85) which might indicate that different physical mechanisms play a role in the high emission scenario. For the CMIP5 ensemble it is also the case for periods ending after 2030 but not before. In the CMIP6 ensemble there is a sharp increase correlation around 2010 for all period length. This could be because the AMOC in CMIP6 models is dominated by internal variability up until around 1990 after which a sharp decline sets in (Fox-Kemper et al. 2021). It could take about 20 years for the AMOC decline to start dominating ODSL changes in the North Sea. We want to select a period that overlaps with the altimetry period that started in 1993, that is as long as possible to reduce the influence of natural variability on the estimation of the trend and that shows significant correlation between past trend and end of century height for both CMIP5 and CMIP6 and for all emission scenarios. We find that for both model ensembles the rate over the full satellite altimetry period 1993-2021 with a length of 29 years provides reasonably high correlation coefficients. For this period the correlation coefficients are 0.54, 0.57, 0.61 respectively for RCP2.6, RCP4.5, RCP8.5 and 0.63, 0.57, 0.42 for SSP1-2.6, SSP2-4.5 and SSP5-85. Those coefficients are all significant, e.g. the null hypothesis of no-correlation is rejected with a p-value between 0.0005 and 0.03. This period is therefore selected for further analysis.

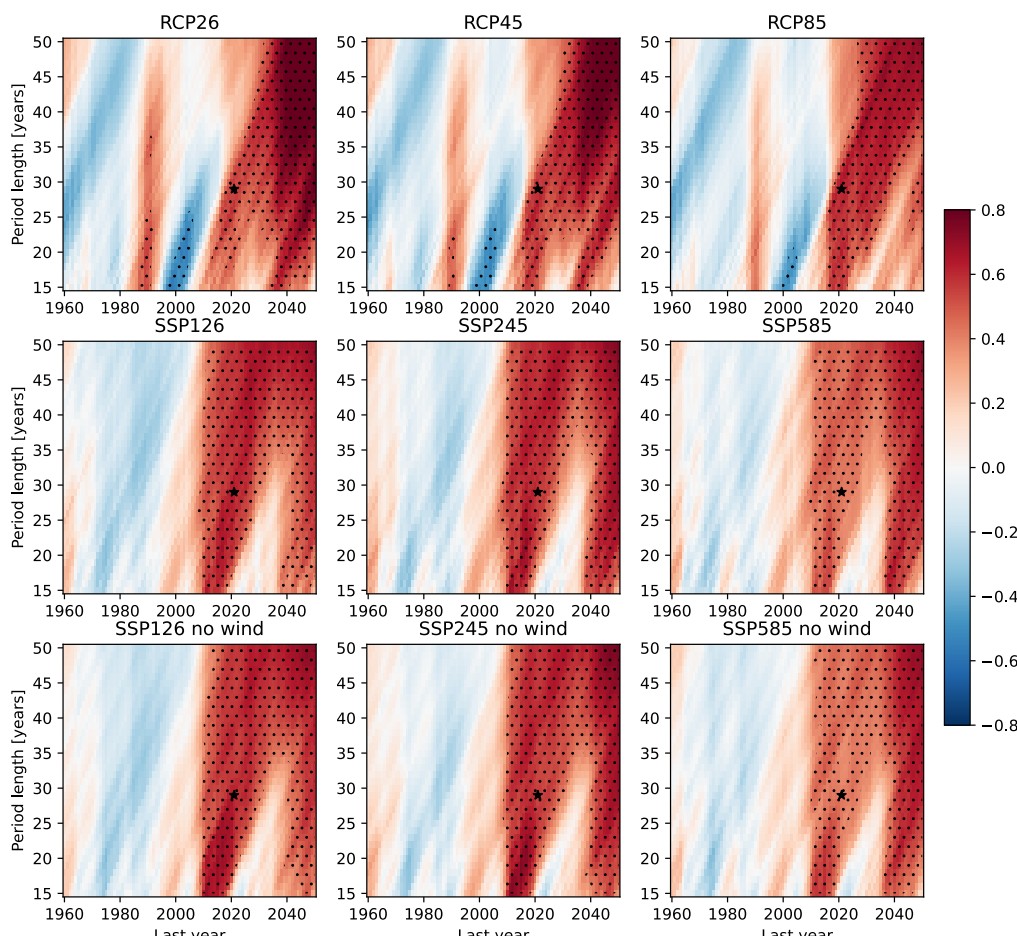

**Figure 2: Pearson correlation coefficients between rate of ODSL computed for different periods before 2050 and average height anomaly in the period 2090-2099. The columns represent different emission scenarios, and the rows show CMIP5, CMIP6 and CMIP6 with wind corrected, respectively. The last year of the period used to compute the rate varies with the x-axis and the total period length varies with the y-axis. The black star in all panels represent the period ending in 2021 with a length of 29 years which is 1993-2021. Statistically significant values (p-value less than 0.05) are stippled.**

We define three observationally based estimates of ODSL to select the best CMIP6 models for sea-level scenarios. We compute ODSL as the difference between observations and the sum of the other known contributions to sea-level change. We apply this method to a relative sea-level budget based on the measurement from 6 tide gauges and to a geocentric sea-level budget based on satellite altimetry region along the coast of the Netherlands (Fig. 1b). These two budgets, even though they are based on different observations, provide similar estimates of ODSL trend over the period 1993-2021: $0.8 \pm 0.3$ and

0.7 ± 0.4 mm/yr respectively for the tide gauge and altimetry budgets (red dots in Fig. 3). We now assume that the regional steric sea-level change in the deep ocean around the continental shelf makes its way onto the shelf. Using two different regions, two different gridded observational products, and integrating steric anomalies down to a depth of 2000 m we obtain 4 estimates of ODSL change (green dots in Fig. 3). The lowest estimate is obtained from EN4 in the extended Bay of Biscay

(0.1 ±0.3 mm/yr) and the highest is obtained from EN4 in the Norwegian Sea (0.7 ±0.3 mm/yr). To avoid the strict assumption that steric sea-level anomalies in the deep ocean close to the shelf have a direct influence on the shelf, we also use an ocean reanalysis. The SODA reanalysis provides a trend of -0.1 ± 0.4 mm/yr for the period 1993-2020 (yellow dot in Fig. 3).

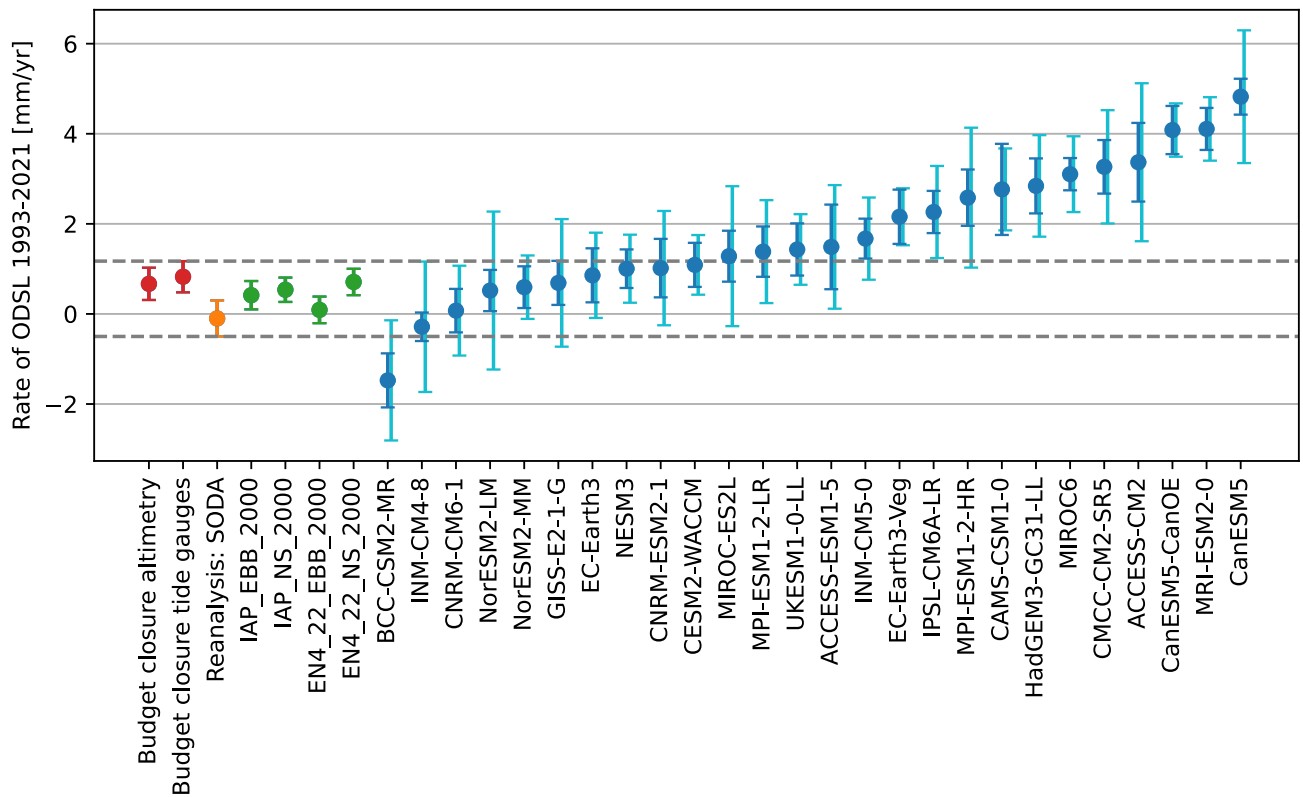

**Figure 3: Rate of ODSL over the period 1993-2021 obtained from different observational methods and CMIP6 models after wind correction (blue). The slopes for CMIP6 models are computed from the historical experiment up to 2014 and SSP2-4.5 from 2015 to 2021. Budget closure based on tide gauge and satellite altimetry data (red), SODA ocean reanalysis (yellow), steric sea level in the top 2000 meters of the ocean from two different temperature and salt**

**databases (IAP, EN4) and two different regions (EBB, NS, see Figure 1). The horizontal dashed lines represent the upper and lower values from observational estimates used to select models. The uncertainty ranges show ± one**

**standard deviation in the estimation of the rate by fitting a linear trend to the data. For the CMIP6 models, an additional uncertainty range is shown, also ± one standard deviation, but based on the natural variability in the historical simulations.**

We also compute the ODSL trend from CMIP6 climate models for the average of the 6 grid boxes shown in Fig. 1b. The wind influence on sea level is removed before computing the rate to reduce the influence of natural climate variability. The rate of ODSL change for the period 1993-2021 goes from -1.5 mm/yr for BCC-CSM2-MR to 4.8 mm/yr for CanESM5. Even after removing wind influence on sea level and considering a long period of 29 years, part of this broad range might be due

to natural climate variability. The Atlantic Multidecadal Variability could play a role for example (Frankcombe and Dijkstra 2009). However, since we showed that there is a significant correlation between rates of ODSL change over this period and end of the century change, the range is also determined by specific sensitivity of the ODSL to climate change in those models. We now select models with a realistic ODSL rate. We define a broad observational range of realistic ODSL rate that goes from the lowest observational estimate minus one standard error (e.g. -0.5 mm/yr) up to the highest plus one standard

error (e.g. 1.2 mm/yr). Using this observational range, we define 4 model selections. From the strictest (selection 1) to no selection at all (selection 4) (Table 1). For the first selection, we require that a model rate falls in the observational range without taking the uncertainty in the model rate into account. For the second selection, we consider the uncertainty in fitting a linear trend to the model data (dark blue range in Fig. 3) and we require that the uncertainty range of the model rate has some overlap with the observational range. For the third selection, we use the uncertainty in the rate computed from the

historical simulations. This includes the influence of natural variability. The uncertainty of the rate estimation is larger for this method. The ±1 standard deviation ranges are shown in Fig. 3. To have an even larger uncertainty range, and because using ±1 standard deviation is somewhat arbitrary, we select the ±2 standard deviation range. For selection 4 all models are selected. Those 4 selections provide a broad range of potential choices.

| | Method | # models too low | # models selected | # models too high |
|---|---|---|---|---|
| Selection 1 | Uncertainty in the estimation of model rate not considered | 1 | 9 | 16 |
| Selection 2 | Uncertainty in the linear fit to model time series (±1 standard deviation) | 1 | 13 | 12 |
| Selection 3 | Uncertainty in the historical natural variability (±2 standard deviation) | 0 | 22 | 4 |
| Selection 4 | All models are selected | 0 | 26 | 0 |

**Table 1: Method of selection and number of models with a too low rate, selected and with a too high rate for the 4 methods.**

As the selection becomes less strict more and more models with high rates are selected. This results in the ensemble average having a rate above the observational range for selection 3 and 4 (Fig. 4). For stricter selections the model ensemble average rates are in the high end of the observational range. The selection also has an influence on the ODSL height at the end of the century. For stricter selection the ensemble ranges are narrower, and the ensemble means are lower.

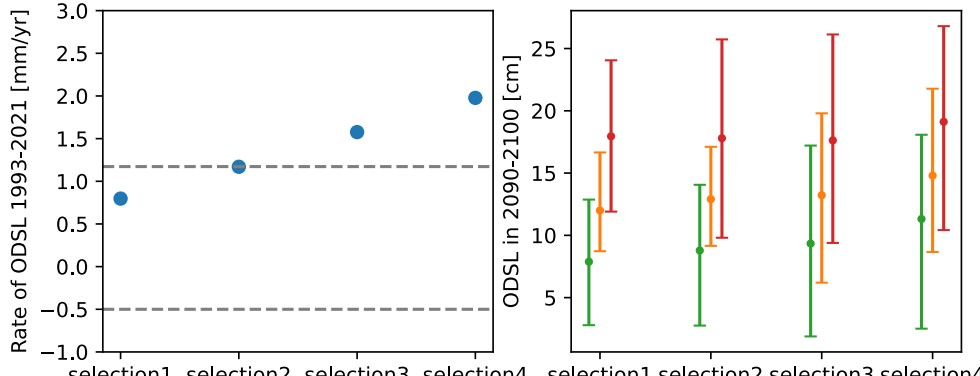

**Figure 4: The left panel shows the ensemble average rate of ODSL for the period 1993-2021 [mm/yr] for the 4 CMIP6 model selections. The horizontal dashed lines represent the upper and lower values from observational estimates. The right panel shows the height of ODSL (mean and 17-83 percentiles range) over the period 2090-2099 compared to the reference period 1986-2005. The three emission scenarios are shown for each selection: SSP1-2.6 (green), SSP2-4.5 (orange) and SSP5-8.5 (red).**

We now compare selection 2 and 4 in more detail with time series (Fig. 5). The difference is larger for SSP1-2.6 than for SSP5-8.5. This is consistent with the fact that for CMIP6 the correlation between rates over the period 1993-2021 and the height at the end of the century is larger for low emission scenarios. For SSP1-2.6 in 2090-2099, the projection for the ensemble of all models is 12 cm with 17th and 83rd percentiles [2-20] while it is 9 [2-14] cm for the model selection. The influence of model selection is especially large for the 83rd percentile.

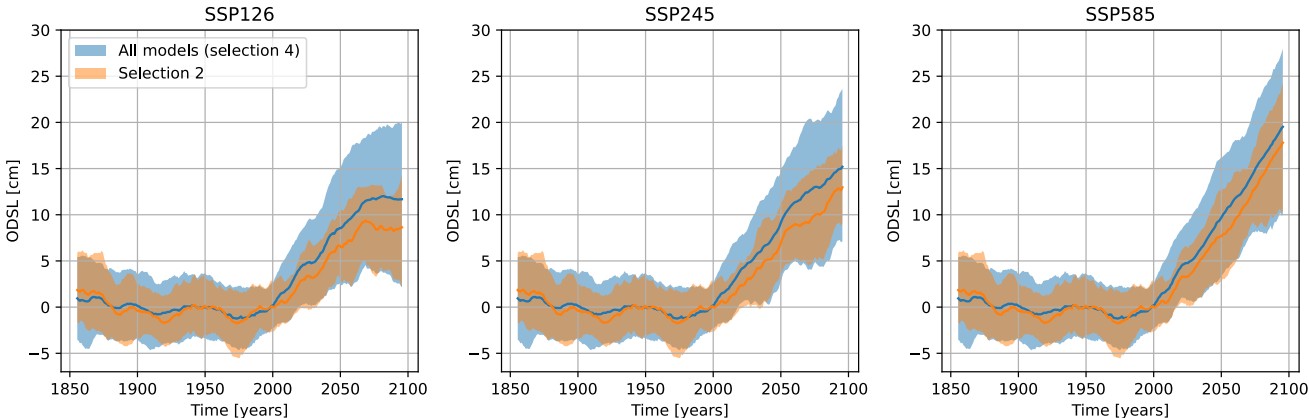

**Figure 5: Time series of ODSL for the full CMIP6 model ensemble (blue) and for the selection of 12 plausible models based on observations (orange). A running average of 10 years is applied. The means and 17th to 83rd percentile ranges are shown.**

**4 Discussion**

While ODSL from CMIP models is used extensively for sea-level projections (Church et al. 2013; Fox-Kemper et al. 2021), there are many limitations that need to be kept in mind. Since a dynamical Greenland ice sheet is not yet included in standard AOGCMs, the influence of freshwater from Greenland melt on ODSL changes are not included in our results. Those effects are both direct and indirect. Fresh water is less dense than ocean water and therefore it raises ODSL locally. The important indirect effect is through a slowdown of the AMOC. The combination of both effects was estimated to be around 5cm in the North Sea for the 21st century in the Community Earth System Model (Slangen and Lenaerts 2016). Based on this study, a rough high-end estimate of Greenland melt influence on ODSL for the period 1993-2021 is 0.5 mm/yr. Since many CMIP6 models already overestimate ODSL, adding this effect would bring them even further from the observational range.

Another, physical process missing in the CMIP models is the nodal cycle, with a period of 18.6 years, that was shown to influence steric sea level in the extended Bay of Biscay region and sea level along the western European coast (Bult et al. 2024). Since the period considered here (1993-2021) starts at a low point of the nodal cycle effect on sea level, it could have contributed to 0.16 mm/yr ODSL change in the observations. This is a relatively small influence compared to the broad range of uncertainty that we consider, but including this effect in the CMIP models would make them even further from the observed range.

While we consider the gravitation, rotation and viscoelastic deformation of changes in mass distribution resulting from land ice melt, changes in land water storage and GIA in the sea-level budget, we do not consider the self-attraction and loading effect of ODSL itself. This effect is present in the observational range but not in the CMIP models. It was estimated to be around 10% of ODSL in the North Sea (Richter, Riva, and Drange 2013). Again, this effect is relatively small compared to

the ranges we have investigated in this study but taking it into account would make ODSL changes 10% larger in CMIP models and bring then further from the observations.

The horizontal resolution and related coarse bottom topography of CMIP models is also a limitation. For the North Sea, a particularly important feature is the English Channel. In a downscaling of a model with a closed English Channel, Hermans et al. (2020) showed that ODSL changes are reduced by around 10cm with a proper representation of the Channel. However, in the CMIP6 ensemble there is no difference in the mean ODSL change of models with open and closed English Channel. More research is needed on the influence of potential systematic biases due to the coarse resolution of ocean models part of AOGCMs.

We found a large overestimation of ODSL change in CMIP6 compared to observations over the period 1993-2021. We estimate this overestimation to be about a factor 3, with a large uncertainty, using the center of the observational range and the CMIP6 ensemble mean. A few processes might be contributing to this overestimation. First, there is an overestimation of climate sensitivity in the CMIP6 ensemble mean (Zelinka et al. 2020). Second, the poleward bias of maximum westerly winds in the North Atlantic in CMIP6 models was associated with a larger ODSL rise (Lyu, Zhang, and Church 2020). Third, in CMIP6 models the AMOC reaches a maximum in the 1980s followed by a sharp decrease which was not there in CMIP5 and does not seem to be in proxy reconstructions either (Weijer et al. 2020). This could be due to an overestimated sensitivity to aerosol forcing (Robson et al. 2022). A fourth process can be inferred from the results of Jesse, Le Bars, and Drijfhout (2024). In that study ODSL changes in the North Sea from CMIP models are fitted with a multilinear regression model with global surface air temperature and the AMOC as regressors. On the one hand, we find that CMIP6 models with an overestimated rate of ODSL according to selection 2 have a sensitivity to AMOC change that is two times larger than those that are in the plausible range. One Sverdrup of AMOC decrease at 35ºN results in 2 cm of ODSL rise instead of 1 cm. On the other hand, those models have a sensitivity to global surface air temperature that is half of that of the plausible models, e.g. one degree warming results in 1.4 cm ODSL rise instead of 2.7 cm. The higher sensitivity of ODSL to AMOC in some CMIP models was explained by the location of deep convection in the North Atlantic (Jesse, Le Bars, and Drijfhout 2024). In models with a deep convection mostly in the Greenland Sea, a reduced AMOC will rise sea level in the North Sea more than in models with a deep convection in the Irminger Sea or Labrador Sea. A fifth potential explanation for the overestimation of ODSL by CMIP6 is that internal variability would have played a role in slowing down sea-level rise during the period 1993-2021. This could be argued for the period 1993-2012 based the results from Richter et al. (2017) who showed that CMIP5 models are able to produce the same pattern of ODSL change as observed in the North Atlantic but not necessarily for the same period. So far, no study has shown that it was the case for the longer period 1993-2021.

The large range of ODSL trend from observations (-0.5 to 1.2 mm/yr) also shows the difficulty to estimate this quantity from observations especially for shelf seas. The three methods we used have important limitations. Using ODSL as the unknown in the sea-level budget makes its estimation dependent on being able to quantify all other components of the budget with a good accuracy. Using the steric sea level in the deep ocean relies on the questionable assumption that there is no horizontal pressure gradient between the deep ocean and the shelf. Bingham and Hughes (2012) suggest that choosing regions closer to

the coast and integrating less deep, results in better correlation between steric sea level and coastal sea level at inter-annual time scale. However, we look at a longer time scale, so it is not clear how their conclusion applies to our study. We look at the difference in Fig. A1 between an integration down to 500m and 2000m. The result show that for both IAP and EN4 the choice of integrating down to 500m would result in smaller rate of steric sea level rise and would not change the range we have now for observed ODSL rate. Ocean reanalysis have the potential to solve those issues and be the best tool to diagnose ODSL along the shelf but still suffer from issues like drifts and biases due to air–sea fluxes, ocean mixing errors, coarse model resolution or the assimilation of observations (Dangendorf et al. 2021).

Given the complexity of ODSL changes, the physical limitations and biases of AOGCMs discussed above, and the different goals of projections it is difficult to say which one of the 4 selections presented here should be used. On the one hand we argue that, in the absence of a good understanding of natural variability of sea level in the North Sea, making projections of ODSL with models that are far away from the observational range as in selections 3 and 4 reduces the trust in the projections (Wang et al. 2021). Therefore, we advise most users of sea level projections to use methods 1 or 2. On the other hand, AOGCMs could be selected because of compensating biases, e.g. have the right past ODSL rate for the wrong reason, and therefore not provide better projections. Additionally, for users of sea level projections who are risk averse, the risk of not selecting models that could provide important information about the future and underestimating future ODSL as a result is not acceptable. In that case, selections 3 or 4 would be better suited. In any case, more work needs to be done to understand and evaluate AOGCMs over multidecadal periods, which is less than the typical century time scale they are usually used for.

This study focused on the coast of the Netherlands, which is part of the North Sea. However, the method could be more broadly applied. Given the smoothness of mean ODSL projections from CMIP models, we would expect similar results for the whole Western European coast. For other regions around the world, the results will be different but the method we developed here would also apply. Estimating ODSL from observations would be easier in places with a narrower shelf, the uncertainty related to physical mechanisms transforming the steric sea-level change from the deep ocean to manometric sea-level change on the shelf would be reduced (Bingham and Hughes 2012). The satellite data that we use to estimate ODSL changes is available everywhere around the world but the number of good-quality, continuous tide gauge measurements is exceptional along the Dutch coast.

**5 Conclusion**

To improve projections of ODSL changes for the coast of the Netherlands based on CMIP6 AOGCMs we looked at the potential for the rate of change for past periods to inform about the height at the end of the century. We found that rates computed over the period 1993-2021 correlate significantly with height at the end of the century. However, correlation coefficients are around 0.4 to 0.6, depending on the CMIP version and the emission scenario, showing that different processes also play a role in driving the spread of ODSL height at the end of the century.

We then estimated ODSL change for the period 1993-2021 with three different methods. The first method assumes that ODSL is the difference between observed sea level and the sum of all known contributors to sea-level rise, e.g. ice sheets, glaciers, land water storage, global steric sea level and glacial isostatic adjustment. We applied this method to both relative

sea level from tide gauges and geocentric sea level from satellite altimetry. In the second method we computed regional steric sea-level change in two regions of the deep ocean outside of the North Sea: the extended Bay of Biscay and the Norwegian Sea. In the third method we used sea-level data from an ocean reanalysis that does not assimilate satellite altimetry data. These three methods provide 7 estimates of ODSL rates of change during the period 1993-2021. Based on these estimates we defined a broad range of plausible values: [-0.5, 1.2] mm/yr.

This range was compared to the rates of CMIP6 models from which the influence of local wind variability was removed. We found that 4 to 16 models simulate a rate that falls above this range depending on how the uncertainty in the rate of AOGCM is considered. We define 4 selection methods going from a very strict method that requires that the rate of an AOGCM falls in the observational range without considering the uncertainty in the rate estimation (selection 1) to using all AOGCMs (selection 4). Selecting models results in lower ensemble mean and narrower uncertainty ranges at the end of the century.

The difference is largest for the low emission scenario SSP1-2.6 for which the median and 83$^{rd}$ percentiles are reduced by about 25% in selection 2 compared to using all AOGCMs. We argue that using a strict selection method (selection 1 or 2) is better for most users except for risk averse users who could consider selections 3 or 4. We discussed a few reasons that could explain the large overestimation of many models. An overestimation of the sensitivity of the local ODSL changes to AMOC changes seem to be playing an important role.

 **Appendix A: Steric sea level change**

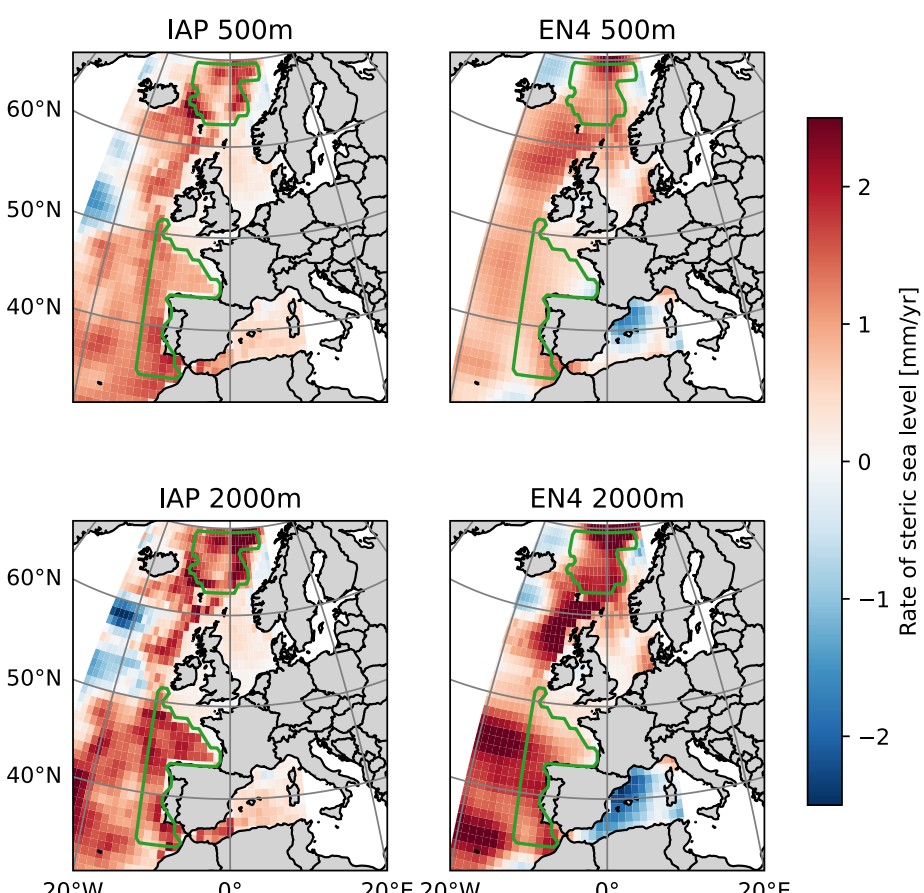

**Figure A1: Maps of rate of vertically integrated steric sea level change in mm per year from the surface to 500m depth (upper row) and from the surface to 2000m depth (lower row). Left column is for IAP and right column for EN4. Green contours indicate the Norwegian Sea and the Extended Bay of Biscay regions.**

**Code and data availability**

The EN.4.2.2 data were obtained from https://www.metoffice.gov.uk/hadobs/en4/ and are British Crown Copyright, Met Office, provided under a Non-Commercial Government Licence http://www.nationalarchives.gov.uk/doc/non-commercial-government-licence/version/2/. To compute ocean density we used the GSW-Python toolbox https://teos-10.github.io/GSW-Python/. The budget data from Frederikse et al. (2020) can be downloaded at https://zenodo.org/record/3862995. The ICE6G

data is available at https://www.atmosp.physics.utoronto.ca/~peltier/data.php. The ERA5 data is available from the climate data store https://cds.climate.copernicus.eu/cdsapp#!/dataset/reanalysis-era5-single-levels-monthly-means?tab=overview.

The code developed to compute the sea-level budget is available on Github https://github.com/dlebars/SLBudget. The SODA 3.4.2 data can be found here https://dsrs.atmos.umd.edu/DATA/soda3.4.2/REGRIDED/ocean/. The code developed to analyse the SODA data is available on Github https://github.com/iris-keizer/ROMS-project/tree/main/ROMS-project/local%20notebooks/analysis/SODA. The output from CMIP6 climate models is available from the Earth System Grid Federation (ESGF) CMIP6 search interface (https://esgf-node.ipsl.upmc.fr/search/cmip6-ipsl/). The code developed to compute the wind influence on CMIP6 ODSL is available on Github https://github.com/iris-keizer/Thesis-KNMI/blob/main/Wind_contribution/Analysis/notebooks/nearby_wind_regression_cmip6.ipynb. The code developed for the data analysis and figure production is available on Github code https://github.com/KNMI-sealevel/Obs_ODSL_Netherlands under the GPL-3.0 license.

**Author contributions**

DLB and SD conceptualized the study. DLB and IK performed the data analysis. DLB wrote the first draft. All authors contributed to the preparation of the final draft.

**Competing interests**

The authors declare that they have no conflict of interest.

**Acknowledgements**

We would like to thank Franka Jesse for providing the regression coefficient data from (Jesse, Le Bars, and Drijfhout 2024). This study was part of the development of the new KNMI'23 sea-level scenarios for the coast of the Netherlands.

**Financial support**

This publication was supported by the Knowledge Programme Sea Level Rise which received funding from the Dutch Ministry of Infrastructure and Water Management, and PROTECT, which received funding from the European Union's Horizon 2020 research and innovation program (Grant No. 869304). PROTECT contribution number [TO FILL UP BEFORE PUBLICATION].

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
