# Peer review of "Constraining local ocean dynamic sea-level projections using observations"

_EGUsphere, 2024_

## Author Response (AR1)

**Answers to reviewers**

Constraining local ocean dynamic sea-level projections using observations

Dewi Le Bars, Iris Keizer, and Sybren Drijfhout

**Anonymous referee #1**

The study is motivated by the ocean dynamic sea level (ODSL) change being a major contribution to future coastal sea-level change. Yet, there's a large spread in CMIP projections of this component. The authors present a method to reduce the spread by constraining ODSL changes by the end of the century using observations. They focus on the region off the Dutch coast – the southern part of the relatively shallow North Sea. The authors seek a across-model relationship between ODSL rates over various periods with the ODSL change by the end of the century to identify the historical period that shows most potential to constrain projections with observations.

They then use three different, largely independent, set of observations giving a range of present-day (1993-2021) rates of ODSL change and select models that fall in this range to constrain ODSL projections. Using this approach, they find that the ODSL contribution to sea-level rise using all CMIP models might be overestimated.

The paper is well written, the methods are clear, and the results well presented. The authors discuss a serious of potential, plausible reasons for the overestimation of "conventional" ODSL projections but also acknowledge the limitations of their approach. Those are, in my opinion, to a large degree related to the importance of internal variability at relatively small spatial scales and short-ish (<30 yrs) time scales as well as a lack of understanding the mechanisms that lead to ODSL changes on shallow continental shelves.  The authors have discussed the latter and shown that the observational record may be just long enough to reduce the effect of internal variability on their results (although this could be stated more clearly, see below).

*Authors response:* Thank you for reviewing our work and for your constructive comments. We provide detailed answers to your comments below and indicate how we used them to improve our manuscript. Please note that e.g. an AMOC decline will lead to increase OSDL in the North Sea. This implies that apart from internal variability, models can feature spurious trends when projected circulation changes are overestimated by a too large model sensitivity to climate change, including global warming and aerosol reductions (Robson et al. 2022). We now comment on this in the discussion:

"Third, in CMIP6 models the AMOC reaches a maximum in the 1980s followed by a sharp decrease which was not there in CMIP5 and does not seem to be in proxy reconstructions either (Weijer et al. 2020). This could be due to an overestimated sensitivity to aerosol forcing (Robson et al. 2022)."

And also:

"A fifth potential explanation for the overestimation of ODSL by CMIP6 is that internal variability would have played a role in slowing down sea-level rise during the period 1993-2021. This could be argued for the period 1993-2012 based the results from Richter et al. (2017) who showed that CMIP5 models are able to produce the same pattern of ODSL change as observed in the North Atlantic but not necessarily for the same period. So far, no study has shown that it was the case for the longer period 1993-2021."

**General comments**

The authors acknowledge that the presence of internal variability impacts the results. From Figure 2, it seems like forced rates take over for periods ending after ca 2015 (CMIP5) and 2010 (CMIP6) respectively. Yet at periods up to 30 years (approx. the length of the observational record used here) there is still a potential contribution from internal variability (at least in CMIP6) as shown from the correlations with periods ending 2020-2030. Could the authors comment on how that affects their results? Is there a way to quantify this effect, using for example control simulations to quantify the strength of internal variability, for example using the control simulations?

*Authors response:* To compute the influence of natural variability on the trend we used the historical simulations of each model (see method in "ODSL from CMIP5 and CMIP6 models", in section 2). Those estimates are now provided in figure 3. Additionally, to estimate the influence on the projection we now compare different model selection choices.

I'm also curious about the strong negative correlations in the CMIP5 models for short periods (<20-25 years) ending around 2000. Are they significant? If so, what is the explanation.

*Authors response:* We added the statistical significance information to figure 2. Those negative correlations are significant but we do not have an explanation for them. We see that the average trends over those periods (not shown in manuscript) are themselves negative or close to zero. Aerosols forcing could have played a role but we prefer not to speculate in the manuscript and draw the conclusion that studying longer periods, for which the behaviour of both CMIP5 and CMIP6 ensembles is more consistent, is better.

Though not the focus of this study I noticed the large range the observational estimates cover. Some of them don't even seem to overlap. I think this could be discussed a bit more.

*Authors response:* We added this paragraph in the discussion:
"The large range of ODSL trend from observations (-0.5 to 1.2 mm/yr) also shows the difficulty to estimate this quantity from observations especially for shelf seas. The three methods we used have important limitations. Using ODSL as the unknown in the sea-level budget makes it estimation dependent on being able to quantify all other components of the budget with a good accuracy. Using the steric sea level in the deep ocean relies on the questionable assumption that there is no horizontal pressure gradient between the deep ocean and the shelf. Bingham and Hughes (2012) suggest that choosing regions closer to the coast and integrating less deep, results in better correlation between steric sea level and coastal sea level at inter-annual time scale. However, we look at longer time scale, so it is not clear how their conclusion applies to our study. We look at the difference in Fig. A1 between an integration down to 500m and 2000m. The result show that for both IAP and EN4 the choice of integrating down to 500m would result in smaller rate of steric sea level rise and would not change the range we have now for observed ODSL rate. Ocean reanalysis have the potential to solve those issues and be the best tool to diagnose ODSL along the shelf but still suffer from issues like drifts and biases due to air–sea fluxes, ocean mixing errors, coarse model resolution or the assimilation of observations (Dangendorf et al. 2021)."

**Minor comments**

Decide whether you want to hyphenate sea level whenever it precedes a noun (e.g. sea-level projections, sea-level variability etc).

*Authors response:* We now hyphenate sea level when it precedes a noun consistently.

ODSL from CMIP models, post-processing: Should you not first remove the drift from the models and then remove the global mean?

*Authors response:* You are right and this is what we do. There was an error in the method description which we now fixed.

Line 114: "remove it to the ..." -> remove it from the....
*Authors response:* Corrected.

Line 120: Out of curiosity: why did you remove the influence of the wind only for CMIP6 and not CMIP5 models?

*Authors response:* The process requires downloading additional model variables. Given that it does not seem important for CMIP6 we did not do it for CMIP5.

Figure 2: Are all the correlations shown statistically significant? If not, could you show in the plot which correlations are significant?

*Authors response:* We added this information to the figure.

Paragraph starting Line 168: I think there is quite a bit of repetition here. You explained already how you arrived at the observational estimates in Data and Methods. Thus, this paragraph can be shortened.

*Authors response:* We shortened those paragraphs and move some content to the data and method section.

Figure 3, x-labels: In the caption and in Figure 1, Nordic Seas are abbreviated NS. In the x-labels they appear as NWS?

*Authors response:* We modified the x-label of Figure 3 for consistency.

Line 203: I count 13 models that overlap with the observational range and 12 that have a too high rate. Please check.

*Authors response:* This is right, we had made a mistake in the text that is now corrected.

Line 277 Nort Sea -> North Sea

*Authors response:* Corrected.

**Anonymous referee #2**

I have read the manuscript "Constraining local ocean dynamic sea level projections using observations" by Dewi Le Bars et al., submitted for publication to Ocean Science. The manuscript deals with the important issue of model selection for more reliable future climate projections in the context of regional sea level, with a focus on the coast of the Netherlands in the North Sea. The authors begin by identifying, within CMIP5 and CMIP simulations, time periods in which a statistical relationship (i.e., a correlation) exists between past rates of dynamic sea level (DSL) and projected changes for 2090-2099. They identify 1993-2021 as one of such periods and use it as a reference period for selecting and discarding models based on the degree of agreement between observed and simulated rates of DSL in this period. Following this, they produce new DSL projections based on the selected models and show that the model spread is greatly reduced in their new projections.

I do think that coming up with smart and justified ways of selecting climate models is an important component of current efforts to increase our confidence in future projections, and this study represents a contribution towards achieving this goal. Overall, I think this study likely has the potential to be suitable for publication in Ocean Science. The topic at hand is timely and relevant, the paper is well-written, and the

results are adequately presented. My main criticism is that the robustness of the criteria used for model selection is not sufficiently tested and there is little critical discussion on the adequacy of choosing climate models only based on their agreement with observations. Without this robustness analysis and discussion, the claim made in the abstract that "this model selection is better than using all models to provide sea level projections" is not convincing enough, in my view. Below I expand on my concerns and provide suggestions with the hope that they will be useful to the authors in revising the paper.

*Authors response:* Thank you for your careful reading of our manuscript, your postitive evaluation of our work and your criticisms that helped improve the manuscript. We respond to your comments below.

Main concerns:

**1. Robustness analysis**. While I agree that there is no definitive way to prove that the new projections are more accurate than the full ensemble, there are several analyses that can provide clues on the robustness of the results and the adequacy of the approach taken here. I summarize some of those below:

*Authors response:* We agree about the importance to evaluate the robustness of our method. We now added a new Figure 4 and have a clearer rational for our choices in the text. The robustness focusses on the amount of uncertainty to include in the model selection.

The choice of the period 1993-2021 for the selection of the models is somehow arbitrary. How sensitive are the results to the choice of the period? Does the set of selected models change if you use the period 2005-2019 (15 years) instead of 1993-2021? What about if you use the period 1992-2011 (20 years)? All those periods seem to be just as valid as 1993-2021 based on the correlations shown in Figure 2.

*Authors response:* We now motivate our choice of period in a better way. This choice is also clearer now that we have added information about statistical significance in Figure 2. We now write:
"We want to select a period that overlaps with the altimetry period that started in 1993, that is as long as possible to reduce the influence of natural variability on the estimation of the trend and that shows significant correlation between past trend and end of century height for both CMIP5 and CMIP6 and for all emission scenarios"

To what extent do the observed rates of DSL change in 1993-2021 reflect internal climate variability? This is only a 29-year period and thus internal variability might have very well played an important role. If this were the case, then choosing the models that best match observations might not be the best strategy. Could the authors explore this

issue, for example, using an initial condition ensemble under historical forcing to see how large the influence of internal variability is?

*Authors response:* This is right, internal variability still plays a role for trends of 29 years long. We computed this influence based on the historical simulations and added the results to Figure 3. To include this inflormation in the selection of models we now define 4 different model selections. We provide more information about this below.

To estimate the steric sea-level change from hydrographic data the authors vertically integrate the density of sea water from the sea surface down to 2000 m. They refer to Bingham and Hughes (2012) to justify this choice, however, in that paper it was shown that the assumption of no horizontal gradients in sea level (essentially what is assumed here) only works well in the absence of boundary currents. Wouldn't calculating the steric height closer to the coast, for example along the 500-m isobath, be more adequate given the presence of the Norwegian Current? How sensitive is the steric calculation to the depth used in the integration?

*Authors response:* We added a map of steric sea level rate in the appendix (Fig. A1) and discuss this point:

"Using the steric sea level in the deep ocean relies on the questionable assumption that there is no horizontal pressure gradient between the deep ocean and the shelf. Bingham and Hughes (2012) suggest that choosing regions closer to the coast and integrating less deep, results in better correlation between steric sea level and coastal sea level at inter-annual time scale. However, we look at a longer time scale, so it is not clear how their conclusion applies to our study. We look at the difference in Figure A1 between an integration down to 500m and 2000m. The result show that for both IAP and EN4 the choice of integrating down to 500m would result in smaller rate of steric sea level rise and would not change the range we have now for observed ODSL rate."

**2. Adequacy of the selection criteria**. I find Figure 1c to be both interesting and somehow concerning. The ensemble spread is quite narrow during the whole observational period (it is very narrow in the period 1990-2005) and starts growing rapidly in the unobserved period. To me, this behavior indicates that: 1) CMIP models have been strongly tuned to reproduce the observed climate; 2) this tuning process leads to a set of model parameter values that differs from one model to another. That is, different parameter values can lead to models that all agree well with the observed DSL rates, but they produce different projections, hence the large ensemble spread in the future. This seems to cast doubts on the validity of the strategy of selecting models in terms of how well they agree with the observations. Under this strategy, the more "tunned" models will always be selected, but they are not guaranteed to produce the most credible projections (given the parameter degeneracy). There is also the issue of how well models should agree with observations, given the large uncertainty

surrounding observational rates of DSL and the influence of internal climate variability on the rates. I think that all these issues should be considered in any model selection strategy. At the very least, they should be discussed in some detail. In Section 6, the authors provide a nice discussion on the limitations of the CMIP models, but this discussion focuses primarily on processes that are missing from (or not modeled by) the models rather than on the issues that more directly affect the validity of the model selection criteria.

*Authors response:* We are not aware of any climate modeling center using ODSL changes over time to tune their model. More standard variables are usually tuned like historical temperature trend, radiative forcing or aerosol forcing (Schmidt et al. 2017). The tuned variables might correlate with ODSL in the North Sea, but we expect that if models were tuned to reproduce these observed changes in ODSL they would perform better than what we find here. A possible explanation for this is that simulated trends of ODSL in the North Sea are largely influenced by simulated AMOC trends and the model's sensitivity to greenhouse gas increase and aerosol emissions. These forcing were partly compensating during the last 30 years, so their net effect could be tuned, but this would lead to fast dispersion of the various model timeseries when the imbalance between changes in aerosol forcing and greenhouse gas emissions become larger. As a result, we agree that selecting models that perform best in the past is not a guaranty for better performance in the future. Therefore we change the focus of the manuscript. Instead of presenting only one model selection and arguing that it is an improvement, we compare 4 model selections and discuss the advantages and disadvantages of different choices. We added figure 4 to support the comparison and discussion.

Schmidt, Gavin A., David Bader, Leo J. Donner, Gregory S. Elsaesser, Jean-Christophe Golaz, Cecile Hannay, Andrea Molod, Richard B. Neale, and Suranjana Saha. "Practice and Philosophy of Climate Model Tuning across Six US Modeling Centers." *Geoscientific Model Development* 10, no. 9 (September 1, 2017): 3207–23. https://doi.org/10.5194/gmd-10-3207-2017.

**Specific comments:**

Lines 53-54. A brief explanation of the method of emergent constraints would be helpful here.

*Authors response:* Since the method we use is not emergent constraints we prefer not to explain this method here to avoid confusing the readers.

Ocean Reanalysis. Have sea-level trends from SODA been validated in previous studies? If so, I would suggest including a reference here.

*Authors response:* We are not aware of such a study.

Lines 107-108. What data is used to remove wind influences?

*Authors response:* We use ERA-interim to be consistent with the forcing of the reanalysis. We now make it explicit.

Lines 120-122. Internal climate variability can be also due to remote forcing, besides local winds. Have the authors considered this?

*Authors response:* We now consider this effect using the historical simulation of each model (Fig. 3).

From Figure 2, only periods ending in or after about 2010 show a significant correlation between past rates and future changes. The drop in correlation before and after this year is not gradual but very sharp. Physically, why should rates in the period 1993-2010 be predictive of future changes but not rates in the period 1993-2008? This is a bit concerning. Does the cut-off year coincide with the last year of the historical simulations in CMIP5 and CMIP6 models? Could you please comment on this?

*Authors response:* We do not expect and do not see an influence of the switch between historical and scenario simulations in 2006 for CMIP5 and in 2014 for CMIP6. We added this remark in the results section:

"In the CMIP6 ensemble there is a sharp increase correlation around 2010 for all period length. This could be because the AMOC in CMIP6 models is dominated by internal variability up until around 1990 after which a sharp decline sets in (Fox-Kemper et al. 2021). It could take about 20 years for the AMOC decline to start dominating ODSL changes in the North Sea."

Lines 203-205. What is the correlation between past rates and future changes in the 12 selected models?

*Authors response:* We do not think this correlation would provide useful information in the manuscript. Our argument applies to the full ensemble because we select from the full ensemble.

Lines 253. "To be about…"?

*Authors response:* Corrected.

**Communitee comment 1**

This paper presents an interesting and relevant exercise in model weighting for projecting ocean dynamic sea-level change. The results are well presented and convincing, especially because of the significant correlation between historical and future ocean dynamic sea-level change. The correlations are not that high, though, which makes me wonder how much the historical rates in ocean dynamic sea-level change in the CMIP6 models are influenced by internal climate variability and if models

are not now being excluded partially simply because their variability is not in phase with the variability in the observations.

This could (and in my opinion, should) be checked by analyzing multiple initial-condition members, which are available for at least a few CMIP6 models. Alternatively, if this is not feasible for the authors, the influence of internal variability could also be checked by computing rates in windows with the same length in the (de-drifted) pre-industrial control run of each model. The magnitude of the variance in the historical or pre-industrial rates due to internal variability alone would provide important evidence of the robustness of the method that the authors propose.

*Authors response:* Thank you for this suggestion. We now include an estimate of the influence of natural variability based on the historical simulations.

Some other, more minor suggestions:

- L36-37: I would avoid the term 'mitigate' in this context and suggest to rewrite the sentence along the lines of: "so ODSL trends will be positive in some regions and negative in others"

  *Authors response:* We agree with that remark and changed this sentence to "As a result, it contributes to sea-level rise in some areas and to sea-level drop in other areas."

- L114-115: and on the assumption that the drift is linear, have you checked that? *Authors response:* We did not.

- L118: for clarity, could you perhaps explain why the bilinear regridding does not work for all models?

  *Authors response:* We do not know, we did not investigate this. It might depent on the type of grid used by the ocean models.

- Fig3 caption: would be helpful to repeat how the uncertainty in the historical rates of the CMIP6 models is derived

  *Authors response:* We now give some explanation for both uncertainty ranges.

---

## Author Response (AR2)

**Answers to reviewers**

Constraining local ocean dynamic sea-level projections using observations

Dewi Le Bars, Iris Keizer, and Sybren Drijfhout

We would like to thank the referees for the second round of comments. We answer below in blue.

**Anonymous referee #1**

The authors have addressed all my suggestions thoroughly and I deem the manuscript ready for publication. I only have two technical comments:

Figure 3, caption: There are two uncertainty ranges shown for the ODSL rates computed from CMIP6 models. It is mentioned in the main text and mostly obvious but for completeness please mention which colour corresponds to which uncertainty range. We added this information to the caption.

Figure 5, caption: Do you mean 13 (not 12) plausible models? Thank you, it is indeed 13, we corrected the caption.

**Anonymous referee #2**

I have read the revised manuscript "Constraining local ocean dynamic sea-level projections using observations" by Dewi Le Bars et al., submitted for publication to Ocean Science.

First, I want to thank the authors for their efforts to consider my comments. I am satisfied with their approach to accounting for the influence of internal climate variability, though the results confirm what I said previously that, with such influence being as large as it is, choosing the models that best match observations might not be the best strategy. The authors acknowledge that in the discussion, so that is fine. I also think that the new discussion on model limitations and the challenge of estimating ODSL changes has greatly strengthened the paper. Overall, I support the publication of this paper. The paper is rather inconclusive regarding which model selection method is better, but this is also acknowledged and there is some discussion on it, which adds value to the paper.

That said, there is one question that the authors have somehow evaded. In particular, I asked "how sensitive are the results to the choice of the period?" to which the authors responded "We want to select a period that overlaps with the altimetry period that started in 1993, that is as long as possible". This does not really answer my question.

We have added a paragraph in the discussion section to discuss this point:

"We have used the period 1993-2021 (29 years) to select models. The longest period for which satellite altimetry is available. For our selection 2 we found that 13 models were selected. For shorter periods the number of models selected is larger because the uncertainty in computing the rate of sea-level rise from both observations and models is larger. For the periods 1993-2007 (15 years), 1993-2012 (20 years) and 1993-2017 (25 years), the numbers of models selected are 24, 19, and 15 respectively. Out of a total of 26 models. The model selection also depends on the exact period used. For example, while 15 models are selected for the period 1993-2017, for the period 1997-2021 which is also 25 years only 12 models are selected. One model has a rate that is too low to be selected, the same for both periods. The number of models with a too high rate is 10 for 1993-2017 and 13 for 1997-2021 with 9 overlapping models."

Additionally, I find the response that the sharp drop in correlation before and after 2010 is due to a decline of the AMOC in the CMIP6 models that started in 1990 and took 20 years to show up rather unsatisfying. The authors do not provide much evidence to support this statement and, in fact, if the influence of such decline operated on such long time scales, wouldn't we see a gradual drop in correlation rather than the sharp one that we are seeing? There is also the issue that direct ocean observations of the AMOC, at least at 26 N, do not show such a large decline, which again questions the approach to selecting models based on their agreement with observations. I will leave it to the editor to decide on the relevance of the points I raised here.

There is a sharp increase and decrease of correlation before and after 2010 mostly for short periods which we argue are too much influenced by natural variability to use. For the period of 29 years that we use, the increase and decrease are gradual. This can be seen from the graph attached representing the correlation between rate and height in 2090-2100 (y-axis) as a function of the end of the period of computation of the rate (x-axis).